# Foundation Posteriors for Approximate Probabilistic Inference

**Mike Wu, Noah Goodman**
Department of Computer Science
Stanford University
Stanford, CA 94305
{wumike, ngoodman}@stanford.edu

## Abstract

Probabilistic programs provide an expressive representation language for generative models. Given a probabilistic program, we are interested in the task of posterior inference: estimating a latent variable given a set of observed variables. Existing techniques for inference in probabilistic programs often require choosing many hyper-parameters, are computationally expensive, and/or only work for restricted classes of programs. Here we formulate inference as masked language modeling: given a program, we generate a supervised dataset of variables and assignments, and randomly mask a subset of the assignments. We then train a neural network to unmask the random values, defining an approximate posterior distribution. By optimizing a single neural network across a range of programs we amortize the cost of training, yielding a "foundation" posterior able to do zero-shot inference for new programs. The foundation posterior can also be fine-tuned for a particular program and dataset by optimizing a variational inference objective. We show the efficacy of the approach, zero-shot and fine-tuned, on a benchmark of STAN programs.

## 1 Introduction

The primary goal of probabilistic programming is to enable practitioners from any domain to easily reason about random variables of interest [29, 61]. The main challenge is to build posterior inference algorithms that are both efficient for practical usage and universal – working for any program that might be written. Many probabilistic programming languages (PPLs) have been developed [27, 45, 28, 44, 18, 58, 47, 53, 59, 12, 23, 17, 7, 57] each with inference approaches with strengths and weaknesses. Some require users to assist in the inference process by programmatically specifying conditional independencies [7] or hand-crafting variational distributions [52, 7, 17]. Others place strong restrictions on what kinds of programs can be represented [12]. In this paper, we propose a complementary approach to scale inference for probabilistic programs. By treating a program as a constrained form of language with some additional nomenclature, we show that probabilistic inference can be viewed as masked language modeling [62, 19], a technique popular in natural language processing. Since little is assumed about the program, we need impose few constraints on users upfront. Further, the generality of treating inference as a text prediction problem naturally enables more advanced features, such as plating, with little additional work. We call this approach *masked language inference*, or MLI.

While MLI can be used to solve inference queries for a single program, it becomes more powerful when *meta-amortized* (or twice-amortized) to do inference across different queries *and* programs. Success requires a dense set of programs that cover sufficient diversity for the meta-algorithm to generalize. Since this is not always obtainable, we propose *program augmentations* to enlarge a small dataset of programs. The result of meta-amortized MLI on the augmented programs is a *foundation posterior*: a large neural network trained to do inference across many probabilistic

36th Conference on Neural Information Processing Systems (NeurIPS 2022).

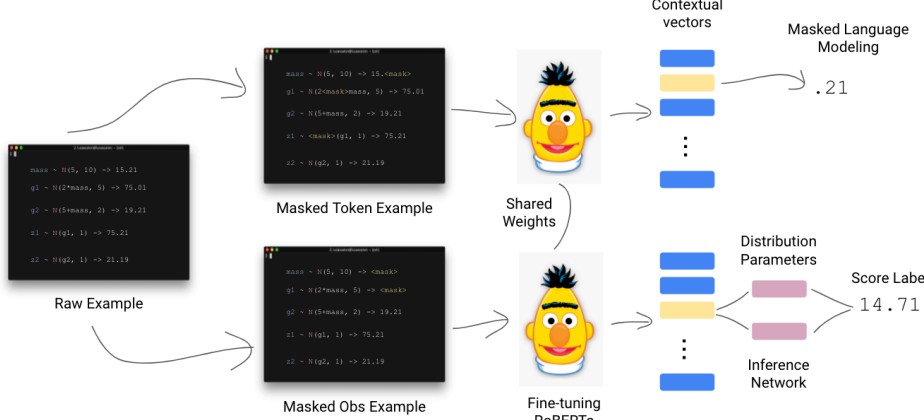

Figure 1: Masked language inference: treating a probabilistic program as a raw string, a large language model is trained to unmask latent variables conditioned on observed ones.

programs. In experiments, we find the foundation posterior to be capable of both zero-shot inference and variational fine-tuning: given a program from the test set, we can achieve higher quality using the foundation posterior as an initial distribution. Indeed, fine-tuning the foundation posterior gives the best performance, for a wide range of runtimes, for a set of standard STAN programs.

## 2  Background

**Probabilistic Programs**    We assume a class of probabilistic programs without loops and such that every line assigns a value to a variable. We note that all finite loops can be "unrolled" into a program without loops. Our constraint primarily disallows probabilistic programs with undecided runtime. Later, we relax this assumption to support loops over conditionally independent variables where unrolling may not be practical. Variable assignment can take many different forms including sampling $x \sim \texttt{gaussian}(z, 1)$, direct assignment $x = z$, and function evaluation $x = \texttt{sqrt}(z)$. Importantly, we do not constrain what distributions and functions are allowed.

**Approximate Inference**    Let $p(x, z)$ be a joint distribution of latent variables $z$ and observed variables $x$. An inference query seeks to compute posterior beliefs $p(z|x) = \left( \frac{p(x,z)}{p(x)} \right)$ which is usually intractable as computing $p(x) = \int_z p(x, z)dz$ faithfully requires solving a difficult integral.

So, we settle for approximate techniques. Markov Chain Monte Carlo (MCMC) [32, 24] and variational inference (VI) [37, 64, 8] are two widely used examples. Focusing on the latter, VI introduces a family of tractable distributions $\mathcal{Q}$ over the latent variables to find the member $q^* \in \mathcal{Q}$ that minimizes the Kullback-Leibler (KL) divergence between itself and the exact posterior $q^*(z) = \arg\min_q \text{KL}\left(q(z)||p(z|x)\right)$. Once found, $q^*$ serves as a proxy for the true posterior.

VI as described above finds $q^*$ for a single observation $x$ but we commonly need to solve multiple inference queries of the same kind but for different values of the observed variables. To amortize [26] the computational cost, we learn a single deterministic mapping $f_\phi : X \to \mathcal{Q}$ that predicts (the parameters of the distribution) $q^*$ as a function of $x$. When scoring a value $z$, we often write the expression $f_\phi(x)(z)$ as a conditional distribution $q_\phi(z|x)$.

To find the function $f_\phi$, we maximize a lower bound of $\mathbf{E}_{p(x)} \log p(x)$ using Jensen's inequality to get $\mathcal{L}_{\text{elbo}}(\phi) = \mathbf{E}_{p(x)}\left[\mathbf{E}_{q_\phi(z|x)} \log \frac{p(x,z)}{q_\phi(z|x)}\right]$ where $p(x)$ is a distribution over the observations.

For probabilistic programs, $p(x, z)$ is easy to sample from (by simply executing the program) but difficult to score. In these cases, we apply a useful trick by optimizing $\text{KL}(p||q)$ rather than $\text{KL}(q||p)$:

$$\mathcal{L}_{\text{comp}}(\phi) = \mathbf{E}_{p(x)}\left[\text{KL}\left(p(z|x)||q_\phi(z|x)\right)\right] = \mathbf{E}_{p(x,z)}\left[-\log q_\phi(z|x)\right] + \text{constant} \qquad (1)$$

Equation 1 is called compiled inference [40]. Dropping the constant, the remaining expression is close to a supervised objective: choose the parameters of the distribution $q(z|x)$ that maximizes the likelihood of $(x, z) \sim p(x, z)$, samples from the probabilistic program.

**Masked Language Modeling** A seemingly unrelated technique is masked language modeling (MLM) [19], used widely by large language models like BERT [19]. The MLM objective is a word prediction task [56]. Given an input sentence, each token has a probability of being replaced by a special mask token. The model then predicts the original token given the rest of the sentence. In slightly more notation, let $x = (t_1, \ldots, t_{i-1}, \texttt{<mask>}, t_{i+1}, \ldots, t_n)$ be a sequence of tokens that form a sentence where the $i$-th token was randomly masked. The label is then just $y = t_i$. (This presentation assumes only one token is masked for simplicity whereas in general, multiple tokens can be masked in a single sentence.) Then, the MLM objective is written as $\mathcal{L}_{\text{mlm}} = \mathbf{E}_{p(x)} \left[ \log p(y|x) \right]$ where $\log p(\cdot|\cdot)$ is cross entropy applied to softmax beliefs over the full vocabulary.

# 3 Masked Language Inference

We now connect approximate inference to MLM. Suppose we are given a single probabilistic program; we wish to do inference, but do not know apriori which variables in the program are observed and which are latent. During evaluation, an inquirer might hand you observations for any subset of the variables and present you with inference queries for the remaining unknown variables. To solve this challenge we aim to find a dataset of observed and latent assignments from which we can train a model to perform inference. We leverage the programitself to generate such a dataset. To do this, we edit the program slightly: For every line (which by design must be a variable assignment), we add an annotation with the value that the variable takes for a single execution of the statement. We do this with the syntax $\texttt{variable} = \texttt{expression} \rightarrow \texttt{value}$. For instance, $x \sim \texttt{gaussian}(0, 1) \rightarrow 0.132$. Unlike most prior PPLs, our program does not include an explicit $\texttt{observe}$ statement. Rather the values annotated on the right side of each arrow is the observation for the variable in the statement. Given a probabilistic program, execute it and annotate each line with assignments. Next, for each line, randomly mask the assignment with some probability. In other words, $x \sim \texttt{gaussian}(0, 1) \rightarrow \texttt{<mask>}$. Save this masked program and the true assignments for all masked variables as a single data point. To generate a dataset, we loop this procedure until we have a sufficiently large corpus. Since the masking decision is made independently for each line, we can theoretically generate every possible permutation of observed and latent variables.

## 3.1 Objective

The dataset of programs as described is not far from a natural language corpus used by BERT. If we could use it to teach a model to unmask assignments of latent variables, then that is tantamount to inference. We formalize this intuition into a procedure we call *masked language inference*, or MLI. See Fig. 1 for an overview. The objective for MLI is a sum of two loss functions: one for traditional MLM and the other to score the true assignment for a latent variable under a posterior distribution parameterized by a neural network. In more detail, given a program $x = (t_1, \ldots t_n)$, we make two versions: for the first $x_{\text{mlm}}$, we do what MLM typically does, choosing random tokens in the program string to mask; for the second, $x_{\text{inf}}$, we do as described above and randomly mask assignments (the values to the right of the $\rightarrow$ symbol per line). Note that the two inputs $x_{\text{mlm}}$ and $x_{\text{inf}}$ have different tokens masked.

We use a transformer network $f_\theta : X \rightarrow \mathbf{R}^d$ that takes a raw string as input where $\theta$ are trainable parameters. We compute $v_{\text{mlm}} = f_\theta(x_{\text{mlm}})$ and $v_{\text{inf}} = f_\theta(x_{\text{inf}})$, both of which are a sequence of contextual vector embeddings $v = (v_1, \ldots v_n)$, one for each token. For each masked index $i$, we perform two actions, one for each input. For MLM, we have a classification head $g_\theta : \mathbf{R}^d \rightarrow |V|$ (e.g. a linear layer) that maps a vector $v_{\text{mlm}}[i]$ to a probability for each token in the vocabulary $V$. For inference, we have an *inference head* $h_\theta : \mathbf{R}^d \rightarrow \mathcal{Q}$ that maps a vector $v_{\text{inf}}[i]$ to an approximate posterior distribution $q(t_i|x_{\text{inf}})$ in the family $\mathcal{Q}$ for the latent variable corresponding to token $t_i$. For continuous variables, a common choice for $\mathcal{Q}$ is the Gaussian family, in which the network $h_\theta$ would return two vectors representing the mean and standard deviation. Alternatively, if the latent variable is binary, we might choose the family $\mathcal{Q}$ to be Bernoulli where $h_\theta$ returns a probability vector. The choice of distribution is flexible as long as scoring is differentiable (though sampling need not be).

| Program | Test Set Evaluation | | Ablation: No MLM | |
|---|---|---|---|---|
| | $\log p(z\|x)$ | $\mathrm{var}\left\{\log\frac{p(x,z)}{q(z\|x)}\right\}$ | $\log p(z\|x)$ | $\mathrm{var}\left\{\log\frac{p(x,z)}{q(z\|x)}\right\}$ |
| Latent | $-1.538_{\pm 0.1}$ | $1.895_{\pm 1.1}$ | $-3.740_{\pm 0.2}$ | $1.059\mathrm{e}4_{\pm 91}$ |
| Clustering | $-3.406_{\pm 0.4}$ | $1.097_{\pm 0.6}$ | $-8.037_{\pm 3.1}$ | $5.415\mathrm{e}3_{\pm 5\mathrm{e}3}$ |
| Hierarchical | $-3.268_{\pm 0.1}$ | $119.2_{\pm 60}$ | $-7.088_{\pm 0.8}$ | $5.162\mathrm{e}9_{\pm 2\mathrm{e}9}$ |
| Multi-level | $-3.359_{\pm 0.5}$ | $131.3_{\pm 31}$ | $-8.363_{\pm 0.7}$ | $3.219\mathrm{e}8_{\pm 2\mathrm{e}8}$ |
| Milky way | $-2.896_{\pm 0.2}$ | $66.09_{\pm 44}$ | $-5.619_{\pm 0.2}$ | $1.147\mathrm{e}6_{\pm 1\mathrm{e}6}$ |
| Rosenbrock | $-1.827_{\pm 0.1}$ | $6.673_{\pm 3.9}$ | $-4.505_{\pm 0.1}$ | $4.252\mathrm{e}5_{\pm 2\mathrm{e}5}$ |

Table 1: Masked Language Inference on a suite of probabilistic programs. For each program, a test set is built using 1 000 new executions with randomly masked assignments. We measure the average quality of inferring these masked values. We show averages over 3 runs.

In summary, the MLI objective is:

$$\mathcal{L}_{\mathrm{mli}}(\theta) = \mathbf{E}_{(x_{\mathrm{mlm}}, x_{\mathrm{inf}}) \sim p(x)} \left[ \log p(t_{\mathrm{mlm}}|x_{\mathrm{mlm}}) + \alpha \cdot \log p(t_{\mathrm{inf}}|x_{\mathrm{inf}}) \right] \tag{2}$$

where $\alpha$ is a weight balancing the two losses, and $t_{\mathrm{mlm}}, t_{\mathrm{inf}}$ are masked tokens. While Equation 2 only masked a single token per loss, in practice we randomly mask 15% of tokens in $x_{\mathrm{mlm}}$, and mask an increasing amount of assignments in $x_{\mathrm{inf}}$ according to a schedule: we begin at 15% but increase this masking probability throughout training to 50%, thereby increasing the difficulty of inference. The MLI objective in Equation 2 can be maximized with stochastic gradient descent.

## 3.2 Orderless Auto-Regressive Decoding

After training, given a probabilistic program with $m$ latent variables $z_1, \ldots z_m$, there are a number of approaches to use the approximate posterior $q$ to perform inference. The simplest approach is to unmask all latent variables at once using $h_\theta$. However, we might believe that inferring the value of $z_1$, for example, provides new information when inferring the value of $z_2$.

Given some ordered queue of latent variables – e.g. $z_1, z_2, \ldots, z_m$ – we can loop $m$ times, each time embedding the program using the transformer $f_\theta$ and unmasking the next latent variable in the queue $z_i$ using the inference head $h_\theta$. Then, we remove $z_i$ from the queue and repeat for $z_{i+1}$. Each iteration of the loop produces a new observation, thus changing the program tokens and the resulting embedding through $f_\theta$. However, our choice of ordering in the queue was arbitrary. Because MLI trains the inference model by randomly masking random variables, any combination of observed and latent variables is within domain. Thus, we are free to choose any random order of variables $z_1$ to $z_m$.

Throughout this auto-regressive decoding, the proportion of latent to observed variables will shrink over time. In the first iteration, there will be the maximum number of latent variables whereas in the last iteration, there will only be one. As such, a properly trained MLI model must handle different numbers of latent variables, not just different variables being observed. In Section 3.1, we described how the percentage of masked assignments is varied in training. A benefit of this design is that both programs with very few and very many latent variables are in-distribution.

## 3.3 Toy Experiments

To build intuition and demonstrate efficacy, we consider six probabilistic programs used in prior work [13]. These programs implement popular models like Gaussian mixture models, hierarchical latent variable models, astronomical models, and incorporate library imports like the Rosenbrock function whose code is not specified in the program. We recommend the reader refer to Section A.3 for details. While these programs are simple, they specify wide prior distributions that result in high sample diversity over repeated executions, which should make the MLI task more challenging.

**Evaluation Metrics**   We build a test set with 1,000 new executions of the program not used in training and randomly mask assignments. This set is held constant across runs and ablations. We evaluate performance with two metrics. First, we compute log probability of unmasked assignment under the approximate posterior specified by the model, averaged over all masked tokens per execution and all test executions. The larger this value is, the better the inference.

Second, we compute the variance of the log importance weights (IW), as in [68]. Since the approximate posterior $q$ acts as an importance distribution estimating the true posterior, we measure the quality by the variance of $\frac{p(x,z)}{q(z|x)}$, where lower variance is more desirable. If $q(z|x) = p(z|x)$, the importance weight would be a fixed constant, meaning zero variance.

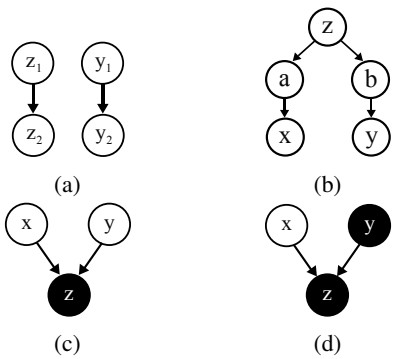

(a)    (b)

(c)    (d)

Figure 2: Graphical model representations for the 'Independent Gaussians' (a), 'Conditional Independence' (b), and 'Common Effect' (c,d) programs.

**Results**    Table 4 reports the performance on the six programs. We observe large probabilities (log close to 0) and small variance of log IW, which suggests good generalization of inference to new sample values. As a baseline, we include an ablation to MLI by removing the MLM term, which amounts to training the inference head only. We find smaller log probabilities (every log point is a significant difference) and much larger variance. Together, this suggests that the MLM term is important for generalization, likely as it helps the neural network understand program structure. For analysis on the distribution of variances, see Section A.3.

### 3.4  Visualizing Attention Maps

Prior works [6, 63, 1, 14] using transformer networks in natural language have re-purposed the attention weights in the later layers as an mechanism to introspect model logic. In a similar vein, we leverage attention weights to approximate a dependency graph between random variables in the problem.

**Datasets**    We study simple probabilistic programs that exhibit independence and conditional independence between random variables. First, in Figure 2(a), $z_1$ and $y_1$ are two independent Gaussian variables. We should expect independence between the sets $\{y_1, y_2\}$ and $\{z_1, z_2\}$. Second, in Figure 2(b), we design a graphical model with a Bernoulli random variable $z$, and pick $a$ and $b$ using `if` statements to be conditionally independent given $z$. We add two Gaussian variables $x$ and $y$ that independently add noise around $a$ and $b$, i.e. $x \sim \mathcal{N}(a, 1)$. Finally, in Figure 2(c,d), we create a common effect model with two potential causes: choose $x$ and $y$ to be independent Bernoulli variables, and set $z$ to be 1 if $x$ `or` $y$ else 0. We expect $p(x|z, y)$ to differ meaningfully from $p(x|z)$.

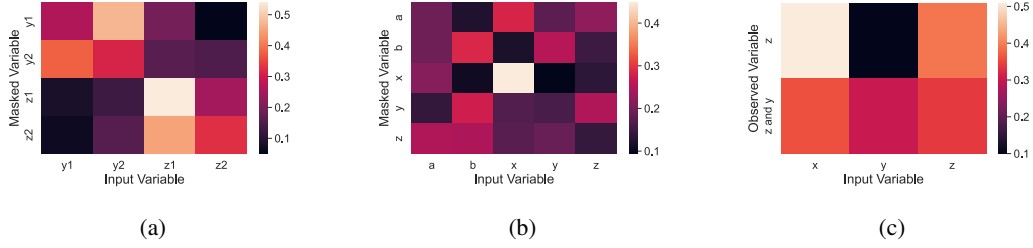

(a)    (b)    (c)

Figure 3: Heatmaps of attention norms for 'Independent Gaussians', 'Conditional Independence', and 'Common Effect' programs. On the y axis, we show which variable has its assignment masked. On the x axis, we list all program variables. The brighter the color, the larger the weight.

**Results**    We report metrics similar to Table 4 for these programs in Section A.4. Here, we instead visualize attention: suppose a single assignment in a program is masked. When inferring its value, we can study the attention weights (obtained from the last transformer layer) on all other tokens in the program. Now, divide the program into lines. Since each line is associated with a variable declaration, we can treat the average weight for all tokens in that line as the "attention" the network is paying to the declared variable when doing inference (for the masked variable). We hypothesize that this measure will be correlated to the graphical dependency between the two variables.

Figure 3 shows the heatmap of attention weights for the three programs. For subplots (a) and (b), we mask each variable's assignment one at a time. The y-axis shows which variable is masked. In the

'Independent Gaussians' program, the model approximates the independence between $\{z_1, z_2\}$ and $\{y_1, y_2\}$. For instance, in the first row of (a), when unmasking $y_1$, we see the network upweights the tokens corresponding to $y_1$ and $y_2$, but downweights the tokens for $z_1$ and $z_2$. The other rows show a consistent pattern. While it may seem peculiar at first for the model to pay attention to $y_1$ when unmasking $y_1$, we note that this is important for the model to understand what kind of variable $y_1$ is. Similarly, in the 'Conditional Independence' program, the model approximates the independence between $\{a, x\}$ and $\{b, y\}$ given $z$. Looking at the first row, we see that unmasking $a$ focuses on variables $x$ and $z$ and downweights $b$ and $y$. Finally, in subplot (c), the y-axis shows which variables are observed rather than which are masked, for clarity. The bottom row plots attention weights if we observe both $z$ and $y$ whereas in the top row, we only observe $z$. We see a stark contrast: in the bottom, the weights are roughly uniform; on top, the model gives little weight to $y$ to unmask $x$.

# 4 Meta-Amortized Inference

Rather than restrict to a single program, we wish to use MLI to do *meta-amortized inference*, or zero-shot inference across programs. Given a dataset of programs for training, how well can we do inference out-of-the-box for a new, unseen program? We assume this novel program is within the same meta-distribution as the training set of programs; otherwise the problem is unsolvable.

## 4.1 Program Augmentations

From prior work [30, 16, 13], we know that the meta-training set must be crafted carefully for meta-amortized inference to perform well: the model must see enough programs that 'span' the program space to be able to generalize to new programs from the meta-distribution. Collecting such a dataset for probabilistic programs is unfeasible. Instead, we propose a set of *program augmentations*, inspired by data augmentations, to enlarge a small set of programs. The hope is that the larger dataset provides enough coverage to generalize to a new test program. We propose the following augmentations family, which can be recursively stacked on each other:

**Fuzz Function** Replaces a primitive function or distribution with a randomly sampled function or distribution. Replacements are constrained to have the same number of arguments as the original function. For example, the program $z = \texttt{rosenbrock}(a, b)$ might be augmented to $z \sim \mathcal{N}(a, b)$.

**Fuzz Constant** Constants are replaced by newly sampled constants from known prior distributions. For example, $r \sim \mathcal{N}(0, 1)$ becomes $r \sim \mathcal{N}(0.1, 0.9)$.

**Line Swaps** Swaps two independent program lines such that the dependency graph between random variables is unchanged. For example, a program with three lines $u \sim \mathcal{N}(0, 1); v \sim \mathcal{N}(1, 1); r = u+v$ could be augmented to $v \sim \mathcal{N}(1, 1); u \sim \mathcal{N}(0, 1); r = u + v$ but it would not be possible to swap the third line with any other given the dependency of $u$ and $v$ on $r$.

**Cut and Glue** Replace the usage of a variable with another variable already defined in the program. Note that this augmentation changes the dependency graph. For example, $u = a + b; v = \texttt{sqrt}(u)$ could become $u = a + b; v = \texttt{sqrt}(b)$ where $b$ is defined earlier in the program.

**Create and Use** Creates a new random variable and uses it in lieu of another variable or constant in the right-hand side of an expression. Note that this augmentation introduces a new random variable. For example, $u = a + b; v = \texttt{sqrt}(u)$ could become $u = a + b; r \sim \mathcal{N}(0, 1); v = \texttt{sqrt}(r)$.

Given that many of these augmentations meaningfully change the program by removing, editing or adding random variables, repeated applications of random augmentations can create novel programs in structure and content. In practice, not all augmentations when compounded result in legitimate programs. For instance, the augmentation $\texttt{rosenbrock}(1, -1) \rightarrow N(1, -1)$ by **fuzz function** is ill-defined. We perform rejection sampling and discard improper programs.

## 4.2 Toy Experiment with Augmentations

To test program augmentations, we revisit the six programs from Section 3.3. However, now we use five of them for meta-training, and hold out the "Rosenbrock" program for meta-test. We apply up to five random program augmentations to each program in both sets to create the training and test splits. Note that the test inputs are all derived from a novel program unseen by the model in training.

This is a much more difficult generalization problem than prior experiments. As an important caveat, we assume knowledge of which external functions might be used. In the 'Rosenbrock' program, we use the `rosenbrock` external function. We assume access to this when augmenting programs in the meta-training set such that `rosenbrock` may appear in training programs.

**Results**  We report results in Table 2. The top row shows the log likelihood and the variance of importance weights for MLI. The second row shows an ablation without augmentations (where we generate a dataset of equivalent size by re-executing programs as we did in Section 3.3). We observe a 100x improvement with augmentations, suggesting better generalization to unseen programs.

## 5  Variational Finetuning

So far we have only studied "zero-shot" inference where an amortized inference model, given a new program, must do inference without any new computational expense. In more realistic scenarios, where test programs can look quite different than the training set, obtaining high quality inference in a zero-shot manner can be challenging. In this case, we propose to finetune our pretrained inference model using stochastic variational inference, or SVI [35, 50, 39]. Due to reparameterization requirements, in this subsection, we focus on inference of continuous random variables only. For discrete random variables, future work can study using the concrete relaxation [42] or the family of REINFORCE-based techniques [65, 46, 60].

More specifically, fix a test program $x^*$ that we would like to do more high quality inference for. Suppose $x^*$ has $n$ observed data points $d_1, \ldots, d_n$, and $m$ latent variables we want to estimate $z_1, \ldots z_m$. Further, we are given $\hat{\theta}$, the parameters obtained from optimizing MLI on a pretraining dataset of programs. Since the composed functions $f \cdot h : X \to \mathcal{Q}$ map a probabilistic program to an approximate posterior, we define the shorthand $q_{\hat{\theta}}(z_{1:m}|d_{1:n}) = h(f(x^*))$.

We can optimize the evidence lower bound, which we recall is:

$$\theta^* = \arg\max_{\theta \in \Theta} \mathcal{L}_{\text{elbo}}(\theta) = \arg\max_{\theta \in \Theta} \mathbf{E}_{z_{1:m} \sim q_\theta(z_{1:m}|d_{1:n})} \left[ \log \frac{p(d_{1:n}, z_{1:m})}{q_\theta(z_{1:m}|d_{1:n})} \right] \tag{3}$$

Since we are not amortizing over programs in this finetuning step, Equation 3 does not contain a second expectation over programs. That is, $z_{1:m}, d_{1:n}$ are the specific variables from $x^*$. We initialize $\theta = \hat{\theta}$ to leverage pretrained weights. Note $p(\ldots)$ has no trainable parameters but acts as a likelihood scaling term for different values of $z_{1:m}$ and thus, cannot be dropped.

**Results**  In Table 2, we include an ablation comparing the generalization of inference with and without fine-tuning. We fine-tune the inference network (from MLI pretraining), optimizing Equation 3 separately for each program for 1,000 iterations. As a baseline, we also finetune the network from a random initialization. We observe that fine-tuning MLI further improves log probability by roughly 2 log points with comparable IW variance. On the other hand, fine-tuning from scratch results in poor performance, highlighting the importance of pretraining.

**Plating**  In many applications of probabilistic inference, we have a dataset of observations that can be thousands of entries or more. We would like to make inference queries given all entries but naive unrolling is unscalable. Instead, we extend MLI to support 'for' loops over conditionally independent variables. To do so, we will implement a form of *minibatching* in our inference algorithm.

Figure 4 shows such a plating example. In the left program, we see 10 observations $d_1, \ldots, d_{10}$ such that $d_i \sim$ `gaussian`$(x, 1)$ where $x$ is outside of the plate. But suppose 10 examples are too

| Model | Test Set Evaluation | |
| --- | --- | --- |
| | $\log p(z|x)$ | $\text{var}\left\{ \log \frac{p(x,z)}{q(z|x)} \right\}$ |
| MLI | $-8.160$ $_{\pm 4.5}$ | $15.10$ $_{\pm 9.6}$ |
| MLI - Augmentations | $-246.6_{\pm 98}$ | $355.9$ $_{\pm 37}$ |
| MLI + Finetuning | $-5.727$ $_{\pm 4.0}$ | $16.06$ $_{\pm 7.2}$ |
| Random + Finetuning | $-42.54$ $_{\pm 89}$ | $784.9$ $_{\pm 242}$ |

Table 2: Meta-Amortized Masked Language Inference over a suite of probabilistic programs. A test set of 1,000 programs are constructed using random augmentations on a held-out program.

many to unroll. In the program on the right, we replace the 'for' loop with a `plate(n)` token where the argument $n$ specifies the number of total iterations. Within the plate, we unroll two randomly subsampled iterations $i \in \{1, 4\}$. For multiple executions of this program, the iterations within the plate change, just like minibatching in neural network training. From the perspective of the transformer, this change does nothing more than add a new token to the vocabulary.

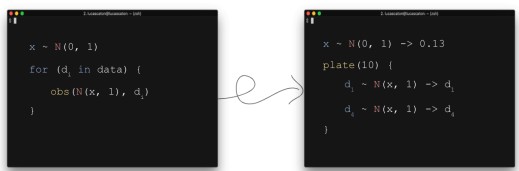

Figure 4: Example of plating within MLI. There is a special `plate` token that specifies the total number of observations, $n$. This defines a scope with a minibatch of $k \ll n$ observations.

**Why does this work?** Consider computing the program density $p(x, d_1, \ldots, d_n)$. This can be rewritten as $p(x)p(d_1, \ldots, d_n|x)$ of which the first term is a given. To compute the second term, observe that $d_1, \ldots, d_n$ are conditionally independent on $x$. So, $p(d_{1:n}|x) = \prod_{i=1}^{n} p(d_i|x)$. Now:

$$\log p(d_{1:n}|x) = \sum_{i=1}^{n} \log p(d_i|x) \approx \frac{n}{k} \sum_{j \in \text{minibatch}} \log p(d_j|x) \tag{4}$$

where the minibatch is of size $k \ll n$. We can make an unbiased (but higher variance) estimate of the full conditional distribution (and hence, the program density) using a small minibatch.

In the Appendix (Section A.8), we demonstrate plating on an item response theory (IRT) [21, 31, 51, 66, 67] model, a popular probabilistic program. In the next section, we will leverage plating use MLI on a bank of real world probabilistic programs with thousands of observations each.

## 6  Foundation Posterior

The relationship between MLI and variational fine-tuning is reminiscent of pretraining and downstream tasks, as popularized by self-supervised learning [19, 15]. Inspired by foundation models [9], we propose to frame the result of MLI as a *foundation posterior* that can be finetuned downstream using SVI to perform inference for a wide array of probabilistic programs. The goal is to pay a large one-time cost in training a "general" inference model which may not be adequate for inference in all settings, but can be quickly adapted to new datasets with low cost. To demonstrate the foundation posterior, we meta-amortize inference over a set of standard Stan [12] programs from PosteriorDB [43], a benchmark dataset for evaluating inference algorithms [4, 5, 69, 70, 20].

**Setup** We hold out three programs from PosterioDB for evaluation, and optimize the foundation posterior on the remaining set. (Programs containing HMMs were removed as we are currently unable to support that graph structure.) Plating with minibatches of size 5 is used for all programs to fit observations within the transformer's 512 token limit. After pretraining, we optimize Equation 4 for each test program individually, varying the number of steps of fine-tuning across 0 (zero-shot), 10, 100, and 1000. As baselines, we use `CmdStanPy` to run Stan-native NUTS and ADVI. For NUTS, we vary the number of mixing steps between 10 and 10k; for ADVI we vary the number of iterations between 100 and 1M. Each program comes with 10k pre-computed posterior samples fit using long runs of expert-tuned NUTS in Stan, constituting the gold standard. To evaluate inference quality we draw 10k samples and perform a hypothesis test between the gold and drawn samples. If the p-value is high, we conclude the two sets of samples are likely from the same distribution.

**Results** Figure 5 reports results on held-out Stan programs. We plot the cost of inference computation (wall-clock time) on the x-axis in log-seconds. For MLI, we sum the cost of fine-tuning and the cost of forward passes to sample. For NUTS and ADVI, we ignore the cost of compilation, measuring mixing time and optimization time, respectively. On the y-axis, we plot the p-value. A better inference algorithm would bias towards the top left corner, achieving a higher p-value at lower

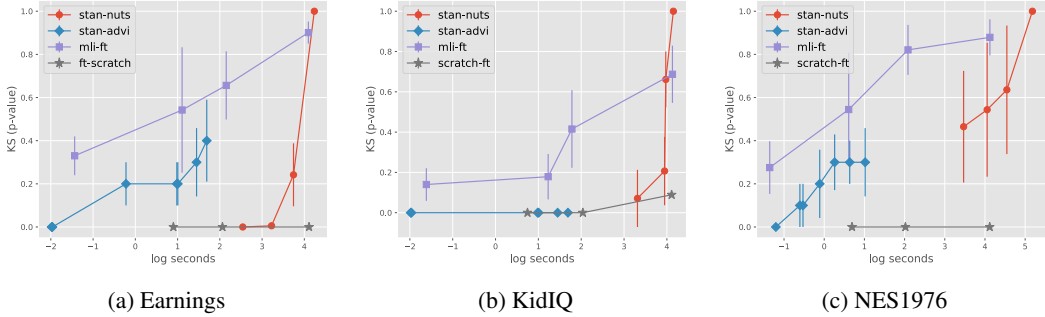

| (a) Earnings | (b) KidIQ | (c) NES1976 |

Figure 5: Comparisons of MLI to native inference methods in the Stan programming language, such as NUTS or ADVI. For any inference algorithm, we draw samples from the posterior and compute similarity to ground-truth posterior samples through a Kolmogorov-Smirnov test.

cost. Each point in the plot represents a setting – we redo inference with different finetuning steps or mixing steps, etc. Each line groups together a single inference algorithm.

First, observe that ADVI in Stan performs poorly (p-value < 0.5) despite being cheap. In the KidIQ program, it is unable to surpass a p-value of 0, suggesting poor posterior samples. Second, we see that NUTS is computationally expensive but converges to a p-value of 1 in all cases, as expected since the gold samples are derived from NUTS. We observe foundation posterior to achieve a compromise between the two: it is cheaper than HMC but achieves higher inference quality than ADVI in Stan. The left-most point in the MLI curve represents zero-shot inference, which we find comparable to ADVI despite the latter training for up to 1M steps. Finally, the gray line shows an additional baseline where we finetune a transformer from scratch. We observe that initializing from a foundation posterior is critical to good inference.

## 7 Related Work

**Amortized Inference** Traditionally, an inference query $q(z)$ is solved for a single assignment of the observed variable $x$. However, in many applications, we may be interested in solving the same inference query for many observations $x_1, x_2, \ldots, x_n$. Re-solving the same inference query from scratch for all $n$ related queries seems wasteful. Amortized inference [26, 55] was proposed as a more efficient alternative by learning a function $q(z|x)$ that maps a observed assignment $x$ to a distribution over the latent variable $z$. In doing so, we "amortize" the cost of inference with a large one-time cost in defining $q$, after which an inference query can be solved with a single function application. Without amortization, the foundation posterior would not be a very efficient inference algorithm.

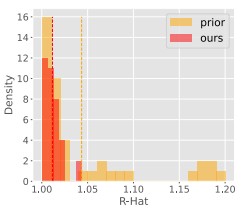

Figure 6: Distribution of R-hat scores over 100 executions of the 'Latent' program when using the approximate posterior (red) versus the prior (orange). Averages are the dotted lines.

**Meta-Inference** Meta-learning has been used to do inference amortized over both queries and a family of generative models [16, 30, 36]. These works have shown that "meta-amortized" inference can generalize well within the family of models it was trained on, even members of the family not explicitly seen in training. The foundation posterior is a meta-amortized algorithm trained with masked language inference. Unlike prior works, we not only evaluate zero-shot inference but also the ability to serve as an effective starting point for further fine-tuning.

Most relevant to our work is a meta-learning approach [13] that builds a white-box inference algorithm by matching every step in a probabilistic program with neural network whose role is to "invert" it. Foundation posteriors trade explainability for flexibility. Our approach is not white-box: inference is performed by a single black box neural network. In return, our approach assumes no structure: it treats the program as nothing more than a string, making its generalization capacity greater than the approach in [13]. We view these two methods as optimizing for distinct properties.

**Foundation Models**    Foundation models [9] are a blanket term for large unsupervised models that map data to representations. Examples include ResNet [33] in computer vision, BERT [19] and GPT [10, 49] in natural language, Wav2Vec [54, 3] in speech, and CLIP [48] and Data2Vec [2] in multimodal applications. Foundation models are treated as frozen backbones whereupon a small portion is finetuned for a downstream task. Our motivation for the foundation posterior is precisely this: learn a good posterior "initialization" that can be quickly finetuned for downstream inference.

## 8    Discussion

**Limitations**    First, optimizing transformer networks is computationally costly, requiring accelerated hardware. For practitioners simply interested in a small set of inference queries, it is simpler to use MCMC or VI. Second, MLI as described is not sufficient for all programs: in PosteriorDB, classes of graphical models like HMMs are difficult to represent as text. Third, our approach, being reliant on deep learning, is limited in its explainability. In the case that inference fails, we are unable to reason about the root cause. Finally, the success of foundation models [9] is largely predicated on large datasets with millions of entries. Unfortunately, we do not know of any large collections of probabilistic programs other than PosteriorDB, which in comparison, is of modest scale.

**Future Work**    Future work could investigate foundation posteriors that act as proposal distributions for MCMC-based methods. As a first step, we run Metropolis Hastings on one of the six toy programs, and compute R-hat [25]. We achieve a score of $1.011$ ($\pm 0.009$ over 100 test programs) when using the foundation posterior as the proposal compared to $1.043$ ($\pm 0.059$) when using the prior distribution. Best practice deems a chain sufficiently mixed if R-hat is less than $1.05$. More analysis is needed to better understand the efficacy of foundation posteriors in this context.

**Summary**    We proposed a meta-amortized inference algorithm for probabilistic programs using masked language modeling. With this algorithm, we build a foundation posterior capable of fast zero-shot inference that can also be finetuned for more accuracy. We are optimistic that the generality of the approach, despite its computational burden, can make for practical and scalable inference.

## Acknowledgments and Disclosure of Funding

This work was partly supported by a NSF Expeditions Grant, Award Number (FAIN) 1918771. MW is supported by the Stanford Interdisciplinary Graduate Fellowship as the Karr family fellow.

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
