# A    Appendix

We provide supplemental descriptions, experiments, and analysis below.

## A.1    Potential Applications of Foundation Posteriors

In Section 8, we presented some preliminary results using the foundation posterior as a prior distribution for Metropolis Hastings. Further, in Section 5, we presented an equivalent technique for seeding variational inference. Here, we more broadly motivate the relationship of foundation posteriors to existing inference techniques for potential future directions.

1. As we did with Metropolis Hastings, it is similarly possible to treat the foundation posterior as a prior or proposal distribution for MCMC, HMC, and NUTS. Ideally, a better proposal would reduce the necessary mixing time.

2. Given the contextual embedding of a new probabilistic program, can we predict the mixing time of MCMC/HMC as a downstream transfer task? A dataset can be collected from PosteriorDB with program embeddings as inputs and true mixing times as target labels. This could be practically important to helping practioners properly use sampling-based approximate inference techniques.

3. Similar to the application above, can we predict other design choices for HMC such as step size, learning rate, or mass matrix? There is a potential to abstract away many of the inference hyperparameters and leverage program embeddings to learn good default choices conditioned on the program text.

Given how established VI, MCMC, and NUTS are as inference algorithms, an immediate practical application of foundation posteriors may be as a preprocessing step for inference methods with more theoretical guarantees, as exemplied in the list above.

## A.2    Additional Figures

Figure 7 shows an example of annotating a probabilistic program and randomly masking assigned observations from three different executions.

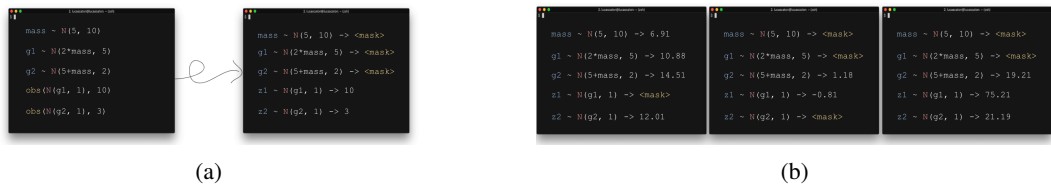

(a)                                                                              (b)

Figure 7: On the left, a standard probabilistic program is reformatted such that `observe` statements are replaced with annotations. On the right are 3 executions with randomly masked variables.

## A.3    Toy Experiments

**Program Descriptions**    **Latent**: Two Gaussian random variables, $\mathcal{N}(\mu_1, \sigma_1^2)$ and $\mathcal{N}(\mu_2, \sigma_2^2)$ where $\mu_2$ is a random affine function of a sample $z_1 \sim \mathcal{N}(\mu_1, \sigma_1^2)$. We choose $\sigma_1 \sim \mathcal{U}(0, 20)$, $\sigma_2 \sim \mathcal{U}(0.5, 10)$.

**Clustering**: Two samples $z_1, z_2 \sim \mathcal{N}(0, 100)$ are divided into two groups $g_1 \sim \mathcal{N}(\mu_1, \sigma_1^2)$ or $g_2 \sim \mathcal{N}(\mu_2, \sigma_2^2)$ using an `if` statement. We choose $\mu_1, \mu_2 \sim \mathcal{N}(-15, 15)$ and $\sigma_1, \sigma_2 \sim \mathcal{U}(0.5, 50)$.

**Hierarchical**: Three random variables $g \sim \mathcal{N}(\mu_g, \sigma_g^2)$, $t \sim \mathcal{N}(g, \sigma^2)$, $z \sim \mathcal{N}(t, \sigma^2)$ each with a mean chosen as a sample from the parent. We choose $\mu_g \sim \mathcal{U}(-5, 5)$, $\sigma_g \sim \mathcal{U}(0, 50)$, $\sigma \sim \mathcal{U}(0, 10)$.

**Multi-level**: Similar to *Hierarchical* but child random variables are modelled as a regression of parent samples where slope and intercept are randomly chosen.

**Milky way**: A probabilistic model for the velocity of two sallite galaxies $v_1 \sim \mathcal{N}(m \times 2, 5)$, $v_2 \sim \mathcal{N}(m + 5, 2)$. The log mass $m$ of the Milky Way is sampled from $N(5, 10)$.

**Rosenbrock**: Computes a noisy Rosenbrock function on samples from two Gaussian variables. The Rosenbrock function is treated as a library import and its code is not provided to the model.

We more thoroughly describe the toy probabilistic programs used in Section 3.3. An equivalent description can be found on pages 17-20 in https://arxiv.org/pdf/2103.00737.pdf.

**Latent** $\mu_1 \sim \mathcal{U}(-5,5); \sigma_1 \sim \mathcal{U}(0,20); z_1 \sim \mathcal{N}(\mu_1, \sigma_1^2); c_1 \sim \mathcal{U}(-33); z_2 = z_1 \times c_1; c_2 \sim \mathcal{U}(-10,10); z_3 = z_2 + c_2; \sigma_2 \sim \mathcal{U}(0.5,10); z_4 \sim \mathcal{N}(z_3, \sigma_2^2)$.

**Clustering** $\mu_1 \sim \mathcal{U}(-15,15); \sigma_1 \sim \mathcal{U}(0.5,50); g_1 \sim \mathcal{N}(\mu_1, \sigma_1^2); \mu_2 \sim \mathcal{U}(-15,15); \sigma_2 \sim \mathcal{U}(0.5,50); g_2 \sim \mathcal{N}(\mu_2, \sigma_2^2); t_1 \sim \mathcal{N}(0,100); m_1 = $ `if` $(t_1 > 0) g_1$ `else` $g_2; \sigma_3 \sim \mathcal{U}(0.5,10); z_1 \sim \mathcal{N}(m_1, \sigma_3^2); t_2 \sim \mathcal{N}(0,100); m_2 = $ `if` $(t_2 > 0) g_1$ `else` $g_2; z_2 \sim \mathcal{N}(m_2, \sigma_3^2)$.

**Hierarchical** $\mu_1 \sim \mathcal{U}(-5,5); \sigma_1 \sim \mathcal{U}(0,50); g \sim \mathcal{N}(\mu_1, \sigma_1^2); \sigma_2 \sim \mathcal{U}(0,10); t_1 \sim \mathcal{N}(g, \sigma_2^2); \sigma_3 \sim \mathcal{U}(0,10); t_2 \sim \mathcal{N}(g, \sigma_3^2); \sigma_4 \sim \mathcal{U}(0.5,10); z_1 \sim \mathcal{N}(t_1, \sigma_4^2); \sigma_5 \sim \mathcal{U}(0.5,10); z_2 \sim \mathcal{N}(t_2, \sigma_5^2)$.

**Multi-Level** $\mu_1 \sim \mathcal{U}(-10,10); \sigma_1 \sim \mathcal{U}(0,100); a_0 \sim \mathcal{N}(\mu_1, \sigma_1^2); \sigma_2 \sim \mathcal{U}(0,10); a_1 \sim \mathcal{N}(a_0, \sigma_2^2); \sigma_3 \sim \mathcal{U}(0,10); a_2 \sim \mathcal{N}(a_0, \sigma_3^2); \mu_2 \sim \mathcal{U}(-5,5); \sigma_4 \sim \mathcal{U}(0,10); b \sim \mathcal{N}(\mu_2, \sigma_4^2); c_1 \sim \mathcal{U}(-5,5); t_1 = b \times c_1; t_2 = a_1 + t_1; \sigma_5 \sim \mathcal{U}(0.5,10); z_1 \sim \mathcal{N}(t_2, \sigma_5^2); c_2 \sim \mathcal{U}(-5,5); t_3 = b \times c_2; t_4 = a_2 + t_3; \sigma_6 \sim \mathcal{U}(0.5,10); z_2 \sim \mathcal{N}(t_4, \sigma_6^2)$.

**Milky Way** $\mu_1 \sim \mathcal{U}(-10,10); \sigma_1 \sim \mathcal{U}(0,30); m_0 \sim \mathcal{N}(\mu_1, \sigma_1^2); c_1 \sim \mathcal{U}(-2,2); m_1 = m_0 \times c_1; \sigma_2 \sim \mathcal{U}(0,10); g_1 \sim \mathcal{N}(m_1, \sigma_2^2); c_2 \sim \mathcal{U}(-5,5); m_2 = m_0 + c_2; \sigma_3 \sim \mathcal{U}(0,10); g_2 \sim \mathcal{N}(m_2, \sigma_3^2); \sigma_4 \sim \mathcal{U}(0.5,10); z_1 \sim \mathcal{N}(g_1, \sigma_4^2); \sigma_5 \sim \mathcal{U}(0.5,10); z_1 \sim \mathcal{N}(g_2, \sigma_5^2)$.

**Rosenbrock** $\mu_1 \sim \mathcal{U}(-8,8); \sigma_1 \sim \mathcal{U}(0,5); z_1 \sim \mathcal{N}(\mu_1, \sigma_1^2); \mu_2 \sim \mathcal{U}(-8,8); \sigma_2 \sim \mathcal{U}(0,5); z_2 \sim \mathcal{N}(\mu_2, \sigma_2^2); r = $ `rosenbrock`$(z_1, z_2); \sigma_3 \sim \mathcal{U}(0.5,10); z_3 \sim \mathcal{N}(r, \sigma_3^2)$.

**Hyperparameters and Training Details** A dataset of $10\,000$ examples are generated by re-executing the probabilistic program. A separate dataset of $1\,000$ examples are generated and held-out in training. For the transformer, we use a RoBERTa [41] architecture, a maximum length of 200, linear learning rate scheduling with $5\,000$ warmup steps, and finetune only the top 6 transformer layers. In the loss, we set $\alpha = 0.1$, the weight on the loss from the inference head. In optimization, we use Adam [38] with a batch size of 16, a learning rate of 4e-3, clip gradients norms at max 1, for 300 epochs. We take the checkpoint with the best loss on a dev set for test evaluation.

**Additional Analysis** We note that for half of the toy programs (especially, hierarchical or multi-level), the variance of log IW is noticeably higher than for the other programs. This pattern persists for the ablation. To study this more, we visualize the histogram of log IWs for the standard MLI and ablation on the 'hierarchical' program in Figure 8. We observe that the majority of the test runs result in log IWs near zero, meaning high quality inference. However, there are a number of examples that the log IW is significantly higher for. The trained MLI model does not generalize as well to these points. Without the MLM loss, we observe these "difficult" cases to be more frequent and severe. In the right subfigure, we ommited points with log IW > 1000 for visibility.

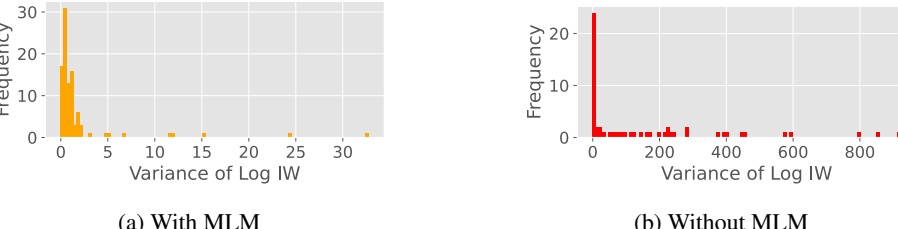

(a) With MLM                    (b) Without MLM

Figure 8: With and without the MLM loss, we see the majority of the log IWs are near zero, suggesting high quality inference. However, there are a number of data points for which the log IW is higher. We observe this to be more frequent and more severe without MLM.

## A.4  Visualizing Attention

We provide more details on the programs and additional results.

**Independent Gaussians**  $\mu_1 \sim \mathcal{U}(-5,0); \sigma_1 \sim \mathcal{U}(0,5); z_1 \sim \mathcal{N}(\mu_1, \sigma_1^2); \mu_2 \sim \mathcal{U}(0,5); \sigma_2 \sim \mathcal{U}(0,5); y_1 \sim \mathcal{N}(\mu_2, \sigma_2^2); z_2 = z_1 \times 2; y_2 = y_1 \times 2$. We expect $z_2 \perp y_2$ and $z_1 \perp y_1$.

**Conditional Independence**  $p \sim \mathcal{U}(0,1); z \sim \text{Bern}(p); a = $ `if` $(z = 1)1$ `else` $10; b = $ `if` $(z=1)3$ `else` $-3; x \sim \mathcal{N}(a,1); y \sim \mathcal{N}(b,1)$. We expect $x \perp y|z, a \perp b|z$.

**Common Effect**  $p_x \sim \mathcal{U}(0,1); p_y \sim \mathcal{U}(0,1); x \sim \text{Bern}(p_x); y \sim \text{Bern}(p_y); z = $ `if` $(x$ `or` $y)1$ `else` $0$. We expect that knowing both $y$ and $z$ should determine $x$.

**Tabular Results**  We include a table reporting log probability and variance of log IW for the three programs above. As these are "simpler" programs than even those in the MLI toy experiments, we observe more favorable results.

| Program | Test Set Evaluation | |
| --- | --- | --- |
| | $\log p(z|x)$ | $\text{var}\left\{\log \frac{p(x,z)}{q(z|x)}\right\}$ |
| Independent Gaussians | 0.856 | 0.779 |
| Conditional Independence | 0.616 | 0.674 |
| Common Effect | 1.215 | 0.889 |

Table 3: Analogous results to Table 4 for the toy problems for visualizing attention.

**Hyperparameters and Training Details**  With the exception of a maximum sequence length of 100, we use the same hyperparameters as in Appendix A.3.

## A.5   Importance Weights

We review how to derive log importance weights. Assume an observed variable $x$ and a latent variable $z$. Fix a realization $x \sim p(x)$ from some data distribution $p$. Then $\log p(x) = \log \sum_z p(x,z) = \log \left( \sum_z q(z|x) \left( \frac{p(x,z)}{q(z|x)} \right) \right) = \log \left( \mathbf{E}_{q(z|x)} \left[ \frac{p(x,z)}{q(z|x)} \right] \right) \geq \mathbf{E}_{q(z|x)} \left[ \log \frac{p(x,z)}{q(z|x)} \right] = \text{IW}$. The rightmost expression is called a log importance weight. We compute the variance of log importance weights by sampling several $z_1, \ldots, z_n \sim q(z|x)$ and computing $\text{Variance}\{\log \frac{p(x,z)}{q(z|x)}, z \in \{z_1, \ldots z_n\}\}$. The better the importance distribution $q$, the smaller the variance. Observe that if $q(z|x) = p(z|x)$, meaning the approximate posterior is indeed the true posterior, then $\text{IW} = \mathbf{E}_{p(z|x)} \left[ \log \frac{p(x,z)}{p(z|x)} \right] = \mathbf{E}_{p(z|x)} \left[ \log p(x) \right] = \log p(x)$, a constant. The variance of a constant independent of $z$ is 0.

## A.6   Additional Results for MLI

In addition to the results in Table 4, we include an additional ablation in Table 4 where the weights of the transformer backbone are initialized using CodeBERT [22] rather than RoBERTa (the default).

| Program | Ablation: CodeBERT | |
| --- | --- | --- |
| | $\log p(z|x)$ | $\text{var}\left\{\log \frac{p(x,z)}{q(z|x)}\right\}$ |
| Latent | $-1.428_{\pm 0.1}$ | $5.01_{\pm 2.7}$ |
| Clustering | $-3.263_{\pm 0.2}$ | $4.962_{\pm 4.3}$ |
| Hierarchical | $-3.691_{\pm 0.0}$ | $30.59_{\pm 17}$ |
| Multi-level | $-3.054_{\pm 0.1}$ | $69.47_{\pm 16}$ |
| Milky way | $-2.571_{\pm 0.0}$ | $38.07_{\pm 34}$ |
| Rosenbrock | $-2.041_{\pm 0.2}$ | $11.36_{\pm 4.8}$ |

Table 4: Ablation of MLI on the six toy probabilistic programs. We initialize the transformer backbone with CodeBERT rather than RoBERTa.

We observe lower variance than with RoBERTa, suggesting that CodeBERT is a viable (perhaps preferrable) alternative. For all STAN experiments, we use CodeBERT.

## A.7 Scientific Notation

We attempted to structure the inference head $h$ to output scientific notation: output two numbers $a \times 10^b$. The potential upside of this design is more resilience to outliers as it requires a smaller change to represent a difference in magnitude of order. However, we found both instability in training as well as worse average performance. This was due to a one-off error in $b$. Note that being off by one unit in $b$ represents an entire magnitude of order. Future work can explore more stable representations of real numbers. For this paper, we opted for the direct representation in lieu of scientific notation.

## A.8 Toy Experiment with Plating

To measure the impact of plating with finetuning, we study a toy experiment with item response theory (IRT) [21, 31, 51, 66, 67], which estimates student ability (a latent variable) using student responses (the observations) to an exam via a generative model. The Rasch model [51] says:

$$p(\texttt{response}_{ij} = 1 | \texttt{ability}_i, \texttt{difficulty}_j) = \texttt{sigmoid}(\texttt{ability}_i - \texttt{difficulty}_j) \quad (5)$$

where $\texttt{ability}_i$ is the ability of the $i$-th student; $\texttt{difficulty}_j$ is the difficulty level of the $j$-th question; and $\texttt{response}_{ij}$ is the binary correctness of the $i$-th student's response to the $j$-th question.

We highlight that both difficulty and ability are unknown, which makes inference hard as many choices of ability and difficulty result in the same difference (and hence probability). In practice, every student answers all the same questions, and each student answers multiple questions. With sufficiently many questions, it is possible to triangulate ability accurately.

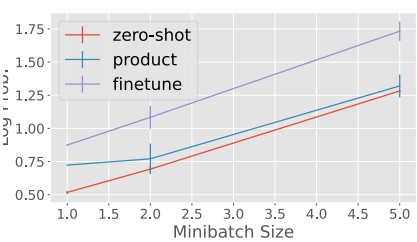

Figure 9: Comparison of finetuning to zero-shot on plated IRT.

To set up the toy experiment, we write a probabilistic program where each student answers 30 questions, where $\texttt{ability}_i \sim \mathcal{N}(0,1)$ and $\texttt{difficulty}_j \sim \mathcal{N}(0,1)$ are both drawn from standard Gaussians [51]. Next, we generate a dataset of 10k programs by ancestral sampling, and optimize Equation 2 (masking random subsets of variables). We do this three times with different minibatch sizes of $k = 1, 2$, and 5 as thirty responses is too long to fit into 512 tokens (for transformer inputs). Separately, we generate 100 test programs not used in training, masking only the ability variable in each.

We study three different ways to do plated inference for ability: the goal is that during test time, we would like to use all student responses (to the 30 questions) rather than just a minibatch of $k$ questions. We compare finetuning with Equation 4 to "zero-shot", a baseline that infers ability using only $k$ observations. Additionally, we include a stronger zero-shot baseline (called "product"): for a test program, we build several minibatches, all of size $k$ but composed of different randomly sampled questions. For each minibatch, perform inference for ability, resulting in a posterior Gaussian. Finally, given several Gaussians, apply a product of experts [34, 68] to arrive at a final posterior. Figure 9 plots $\log p(\texttt{ability}_i | ...$ rest of program $...)$ averaged over test programs for the three approaches. We find that while product improves upon zero-shot performance (by incorporating more information), finetuning achieves higher quality inference still (at the cost of more computation).

**Review of Product of Experts** Suppose we are given $k$ Gausian distirbutions. Then, a product of these Gaussian "experts" is itself Gaussian [11] with mean $\mu = (\sum_i \mu_i \mathrm{T}_i)(\sum_i \mathrm{T}_i)^{-1}$ and covariance $\Sigma = (\sum_i \mathrm{T}_i)^{-1}$, where $\mu_i, \Sigma_i$ are the parameters of the $i$-th Gaussian expert, and $\mathrm{T}_i = \Sigma_i^{-1}$ is the inverse of the covariance (i.e., the precision).

**Tabular Results** We include a table reporting log probability for the three approaches to inferring ability. This shows the same results as in Figure 9.

**Hyperparameters and Training Details** We separate the choices for pretraining (MLI) and finetuning. In pretraining, we use a maximum length of 512 and a maximum gradient norm of 5. All other parameters are identical to Appendix A.3. For the product of experts, we resample minibatches

| Approach | $k$ | $\log p(z\|x)$ |
|---|---|---|
| zero shot | 1 | $0.5174_{\pm 0.0122}$ |
| zero shot | 2 | $0.6930_{\pm 0.0226}$ |
| zero shot | 5 | $1.2824_{\pm 0.0077}$ |
| product | 1 | $0.7231_{\pm 0.0008}$ |
| product | 2 | $0.7707_{\pm 0.1154}$ |
| product | 5 | $1.3201_{\pm 0.0858}$ |
| fientune | 1 | $0.8743_{\pm 0.0069}$ |
| fientune | 2 | $1.0836_{\pm 0.0866}$ |
| fientune | 5 | $1.7331_{\pm 0.0728}$ |

Table 5: Analogous results to Figure 9 for inferring ability from plated IRT models.

10 times. In downstream finetuning, we take gradient steps on the top 6 layers of the transformer, initializezd with pretraining weights. We perform a separate optimization for all 100 test programs with batch size of 4, learning rate 4e-3, for 1000 iterations and no gradient clipping. We maintained the linear learning rate scheduler with $5\,000$ warmup steps. These hyperparameters were taken from the default RoBERTa config in Huggingface.

## A.9  Stan Experiments

We provide a few more details for the Stan experiments. PosteriorDB can be found at https://github.com/stan-dev/posteriordb. We use the python library posteriordb-python. We filter all posteriors by which ones have ground-truth posterior samples. We remove the following programs: arma-arma11, bball_drive_event_0-hmm_drive_0, bball_drive_event_1-hmm_drive_1, garch-garch11, hmm_example-hmm_example, hudson_lynx_hare-lotka_volterra, mcycle_gp-accel_gp, and one_comp_mm_elim_abs-one_comp_mm_elim_abs. We randomly chose 3 programs for the test set: kidiq-kidscore_interaction, earnings-logearn_height, and nes1976-nes. In pretraining, we generate $10\,000$ executions (with randomly masked variables) for each program for a total of $380\,000$ training programs. We randomly choose minibatches of size 5 for programs with more observations than can fit within a 512 token sequence. Different executions would result in different minibatches. We initialize the RoBERTa network from HuggingFace pretrained weights, finetuning the top 6 transformer layers using MLI. We set the weight $\alpha = 0.001$. We use batch size 4, learning rate 4e-3, $5\,000$ warmup steps, gradient clipping of 1 for 50 epochs. In finetuning, we initialize weights from the foundation posterior and separately optimize for each of the three test programs. All optimization choices stay the same, but we optimize the ELBO objective weighted by the ratio between full observation size and minibatch size. We save checkpoints at 10, 100, and 1000 iterations and record wall clock time at each point.

For Stan NUTS, we use 10 chains, thin set to 10, adaptive delta set to 0.8, and vary the number of sampling and warmup iterations as tuples (100, 50), (200, 100), (500, 200), (1000, 500), (2000, 1000), (5000, 2000), (10k, 5k), (20k, 10k), (50k, 10k). For Stan ADVI, we choose a mean field algorithm, 100 ELBO samples, 1 gradient sample, and vary the number of iterations to be amongst 100, 1k, 5k, 10k, 50k, 100k, 500k, 1M.

## A.10  Resources

We used a single Titan X GPU for optimizing all deep learning models, with 4-16 CPU background workers for loading data. NUTS and ADVI experiments were run in the same context but without GPU support. We used the PosteriorDB code base https://github.com/stan-dev/posteriordb for Stan experiments, which has no prohibitive license specified. We used CmdStan-Py for fitting Stan models https://github.com/stan-dev/cmdstanpy which is licensed under new BSD.