# OpenReview forum: "Foundation Posteriors for Approximate Probabilistic Inference"
_NeurIPS.cc/2022/Conference — NeurIPS 2022 Accept_

### Official Review · Reviewer_mMgG · 2022-07-07

**Rating:** 7
**Confidence:** 4
**Soundness:** 3 good
**Presentation:** 3 good
**Contribution:** 3 good

**Summary:**

This paper presents a method for learning a language model that can perform approximate inference in probabilistic programs. Specifically, given a set of probabilistic programs, a dataset of masked, annotated programs is generated and then trained on using a linear combination of a masked language modeling loss and a variational inference loss. An extra head on the language model outputs the parameters of a variational distribution for the current token. This variational distribution can then be used as an approximation of the posterior distribution, and can be further fine-tuned on individual programs to achieve even better performance. Experiments on both toy programs and a real benchmark demonstrate that the learned model provides reasonable posterior probabilities.

**Questions:**

1. How exactly in equation (2) do you get $p(t_{mlm}|x_{mlm})$ and $p(t_{inf}|x_{inf})$ from $g_\theta$ and $h_\theta$, respectively?
2. How does the architecture deal with having multiple types of variable in the program? Are there separate heads $h_\theta$ for each type of variable, given that the parameterization of q(t_i) changes depending on its type?
3. In Table 1, how does the performance change as a function of \alpha? Is the method quite sensitive to this hyperparameter? How do the scores change as \alpha gets smaller and smaller? (i.e., what happens as the MLM loss becomes more dominant, does the method still perform well or does performance fall off as quickly as when the MLM loss is removed?)
4. There’s evidence in the literature that pre-training these architectures on *anything* can be quite beneficial (e.g., Reid et al, 2022). How does this method perform if you pre-train with only the MLM loss on either (a) the programs or (b) some other text data, and then fine-tune on the posterior inference problem?
5. In Section 5, is $p(d_{1:n}, z_{1:m}) difficult to compute, as alluded to in Section 2? How do you compute it here?
6. What happens if you run ADVI for longer in Figure 5?
7. What representation is ADVI using and would it do better with more parameters?
8. How many parameters are used in each of the models in the experiments?

[Reid et al, 2022] Can Wikipedia Help Offline Reinforcement Learning? Machel Reid, Yutaro Yamada, and Shixiang Shane Gu. https://arxiv.org/abs/2201.12122



**Limitations:**

The authors wrote a very nice discussion of the limitations of the proposed approach, which I much appreciated. No potential negative societal impacts were mentioned but it’s not clear to me that there are any immediate such impacts with this work.


**Strengths And Weaknesses:**

**Originality:** To my knowledge, this work is quite original. While language models have been applied to any number of tasks, it seems there are still more places where they can be applied successfully. In this case to posterior inference for probabilistic programs.

**Quality and clarity:** The quality and clarity of the paper are both high. While some additional information should be provided about the inference procedure (see questions), the overall exposition is quite clear, and the contribution and empirical validation is of high quality.

**Significance:** This paper presents an interesting and novel method that I think will be of some interest to the community both in regards to the ability of a language model to learn meaningful representations for posterior inference and a new method for posterior inference that incorporates ideas from modern machine learning (i.e., transformers over large amounts of data). More work is still needed to become a method used for the latter but this paper solidly demonstrates that that work may be possible.

Overall, this paper was well-written and interesting to read. While the approach presented here is a nice step, I wonder if using the proposed method as a prior or proposal for a method with more guarantees would be more useful in general. I see that this is mentioned in the conclusion and Figure 6 and I’d encourage the authors to further explore such directions.

My main complaint with the paper is that I found Section 3.3 to be somewhat poorly executed and explained. Why look at average L2 norms instead of attention weights from the transformer itself? It’s hard to assign much meaning to the particular values of these norms. The writing and explanation could use some editing and rethinking as well to make the results more clear and compelling.

Minor: I think the term “foundation prior” may be more appropriate, particularly if you frame  fine-tuning as an inference procedure that incorporates data into a prior to produce a posterior.

---

> ### Author Response · Authors · 2022-08-02
> **Author Response to Reviewer mMgG**
>
> Thank you for the questions and comments! They encouraged us to add more detail into the revised draft. Below, we hope to address outstanding questions.
>
> > Is the proposed method more interesting as a prior for a more theoretically grounded method?
>
> This is a great point! It could very much be true, in which case a “foundation prior” sounds very reasonable. We have some initial promising results that the approximate posteriors obtained from our approach serve as good proposal distributions for Metropolis Hastings. We believe there is a wealth of directions for future work to study how these distributions can be used for MCMC or HMC. So, it is entirely possible that this approach is much more meaningful practically in the context of other inference methods. It is definitely a priority for future research. In this paper we focussed on VI because we find the parallel to the pretraining-then-finetuning paradigm of language modeling intriguing and constructive.
>
> We added several new potential directions in this vein in Appendix A.1, such as using embeddings to predict mixing time! We encourage the reviewer to read through them.
>
> > How exactly in equation (2) do you get p(t_mlm | x_mlm)  and p(t_inf | x_inf) from gθ and hθ, respectively?
>
> Good question! For p(t_mlm | x_mlm), we do the usual thing in MLM: first, compute all contextual embeddings fθ(x_mlm). Let i be the index for t_mlm. Then compute gθ(fθ(x_mlm)[i]) i.e. pass the i-th context embedding into a classification head. This will return a probability vector of size |vocab|. Finally, compute cross entropy using these predicted probabilities and the true token t_mlm (which is an integer label indexing into vocab).
>
> For p(t_inf | x_inf), we again compute all contextual embeddings fθ(x_inf). Let j be the index for t_inf. Then compute hθ(fθ(x_mlm)[j]). The design of hθ depends on the random variable t_inf. If it is continuous, we choose a Gaussian distribution, in which case hθ returns a mean and standard deviation (two floats). If it is binary, we may choose a binomial, in which case hθ returns a single float representing the weight. If it is multi-class, we may choose to use a Categorical, in which case hθ returns a vector of floats (size equals the number of classes), and normalizes them using a softmax. For this paper, we mostly focus on the continuous case. Here, p(t_inf | x_inf) evaluates the likelihood of the real-valued number t_inf under the inferred Gaussian distribution.
>
> > How does the architecture deal with having multiple types of variables in the program?
>
> For this paper, we almost always restrict ourselves to continuous random variables, in which case we use Gaussian approximate posteriors. This makes it easy to do VI as well. The exception is in Figure 3 where we motivate handling binary variables as well.
>
> A more general answer is that you would need to know if each variable is continuous or discrete. Then, the model should route to the correct head e.g. use h_gaussian if continuous and h_discrete if discrete. There is a lot of nuance here, e.g. how do we handle the number of classes, which follows from how random variables are handled in existing systems (e.g. Stan, Pyro) but is not presented within the current paper.
>
> > In Table 1, how does the performance change as a function of \alpha? Is the method quite sensitive to this hyperparameter? How do the scores change as \alpha gets smaller?
>
> For many of the toy experiments (where it was more reasonable to quickly try many settings), we did a grid search over alpha in the range of 0 to 10, where 0 is a sanity check. We generally found it advantageous to choose alpha to relatively balance the magnitudes of the two terms. For our programs, this was around alpha = 0.001. Choosing alpha to be too large decreased performance, much as the MLM ablation showed poor performance. Choosing alpha to be too small also resulted in poor performance as not enough emphasis is given to the inference term. That being said, the exact choice is not extremely important e.g. 0.01, 0.001, 0.005 all result in around the same performance.

---

> > ### Author Response · Authors · 2022-08-02
> > **Author Response to Reviewer mMgG (continued)**
> >
> > > There’s evidence in the literature that pre-training these architectures on anything can be quite beneficial (e.g., Reid et al, 2022).
> >
> > Interesting! One data point we have pointing to the importance of pretraining on probabilistic programs is in Figure 5. The gray line represents the performance of using a transformer pretrained on NLP data (we also tried a transformer pretrained on python code e.g. CodeBERT). Both of these did not do nearly as well as the transformer pretrained by MLI on PosteriorDB. This at the very least says that Equation 2 is worth optimizing.
> >
> > Also, pretraining on only the MLM objective (on probabilistic programs) is equivalent to alpha = 0, which we found to perform poorly as the parameters in the inference heads were practically untrained, resulting in poor posterior predictions.
> >
> > > What happens if you run ADVI for longer in Figure 5?
> >
> > We ran ADVI for up to 1M iterations, which should be enough for convergence on PosteriorDB problems. Subfigures b and c, we see the performance of ADVI to flatten, suggesting more iterations would not improve performance. For subfigure a, we reran ADVI with 5M iterations, resulting in a KS score centered at 0.37, roughly equivalent to that from 1M steps.
> >
> > > What representation is ADVI using and would it do better with more parameters?
> >
> > We use the default ADVI implementation in STAN. See https://arxiv.org/pdf/1506.03431.pdf for a technical description.
> >
> > > How many parameters are used in each of the models in the experiments?
> >
> > In all experiments for our approach our model has 110M parameters (RoBERTa), of which we optimize 40M (top 6 layers of RoBERTA). When doing ablations (e.g. without MLM in Table 1 & 2 or ft-scratch in Figure 5), we use the same model with the same trainable parameters.

---

> > > ### Author Response · Authors · 2022-08-02
> > > **Author Response to Reviewer mMgG (Updated Attention Results)**
> > >
> > > > Why look at L2 norms instead of attention weights?
> > >
> > > We completely agree that visualizing attention weights is a simpler approach. We redid Figure 3a using attention weights, finding the same relationship (actually a bit cleaner!): weights are larger for related random variables when doing inference; for example, when inferring y1 for the "independent Gaussians" model, weights are higher for variables y1 and y2 than for z1 and z2, the latter two being independent to y1.
> > >
> > > Please see Section A.9 in the appendix where we include the updated heatmap as well as a description. In short, we take the attention weight matrix from the last transformer layer, average over heads, and slice along tokens belonging to the latent variable we are doing inference for, and the random variable we are trying to measure the relationship against. We average again over the sliced matrix to arrive at a single number. For the final revision, we will redo all of Figure 3 using attention weights and move the norm version to the Appendix, or remove entirely. The simplicity of the new approach should improve the text clarity as well.

---

> > > > ### Comment · Reviewer_mMgG · 2022-08-08
> > > > **Response**
> > > >
> > > > Thank you for your responses. I'm happy with the work and believe it should be accepted. I've updated my score to 7.
> > > >
> > > > One other piece of potentially related work that may be worth reading is Chen et al "Towards Learning Universal Hyperparameter Optimizers with Transformers" (https://arxiv.org/abs/2205.13320) which uses a language model to do hyperparameter optimization. Perhaps some of the design choices made there could be beneficial/relevant here as well.

---

> > > > > ### Author Response · Authors · 2022-08-08
> > > > > **Author Response to Reviewer mMgG**
> > > > >
> > > > > Thank you for the response! We will look into that paper more carefully as (from a quick scan) it seems to have interesting ideas that may be applicable.

---

### Official Review · Reviewer_qzPM · 2022-07-11

**Rating:** 7
**Confidence:** 3
**Soundness:** 3 good
**Presentation:** 4 excellent
**Contribution:** 3 good

**Summary:**

This paper has formulated probabilistic inference as masked language modeling. In particular, it proposed a meta-amortized inference algorithm for probabilistic programs where a foundation posterior is constructed and can be ﬁnetuned downstream to perform inference over various programs. The authors demonstrated the efficacy of such an approach on a benchmark of STAN programs.

**Questions:**

- expressivity of the supported programs.
If I understand correctly, the proposed approach can be applied to any probabilistic program written in STAN. Is the if-else statement allowed in your probabilistic programs? In other words, whether or what kind of conditional statements can be supported by your approach? It is appreciated if the author can clearly define the kind of probabilistic programs they can support.

- experiments.
I think the experiment sections throughout the paper are slightly weaker than other parts of the paper. It is a little bit difficult for readers with little background knowledge of STAN and the six programs, in particular, to fully appreciate the scope of the programs the paper is targeting. I would suggest enhancing related discussion both in the main paper and in the supplement.

- computation cost.
Though mentioned by the author that their key motivation for the foundation posterior is precisely this: learn a good posterior “initialization” that can be quickly ﬁnetuned for downstream inference. However, the experiments are mostly done on not too difficult programs. I am wondering what kind of probabilistic problems that do really worth the computational cost of optimizing transformer networks and are struggling to be solved by common PPSs such as STAN, pymc3, or Pyro.

**Strengths And Weaknesses:**

**Originality/Significance:**
Good: It is interesting to see that the authors interpret probabilistic inference from a different angle and bridge ideas from unsupervised learning (language modeling in particular). As discussed in the related work section, I believe this idea is novel and also has solid foundations, which opens up interesting research directions.

**Quality:**
Good: The paper appears to be technically sound.


**Clarity:**
Good: The paper is well organized and clearly written. Adequate intuitions, as well as details of the proposed methods, are both introduced and it is enjoying reading the paper.

**Weaknesses:**
One main concern of the paper is its computation burden. I will explain in detail in the following section.

---

> ### Author Response · Authors · 2022-08-02
> **Author Response to Reviewer qzPM**
>
> Thank you for the questions and comments! We hope to address a few of them below.
>
> > Is the if-else statement allowed in your probabilistic programs?
>
> Yes! If you look at the “Clustering” toy problem (see Appendix), it contains an if statement. Additionally, many of the programs in PosteriorDB contain if statements as well.
>
> > What kind of probabilistic programs do we support?
>
> There is a theoretical limitation to our formulation, which is that we don't handle unbounded stochastic structure. This is very similar to Stan, numPyro, and other practical PPLs. (For bounded stochastic structure we unroll, much as those languages do.) However it means our language is not universal. We'll clarify this in the discussion for the final draft.
>
> In practice our formulation handles conditionals, discrete variables, and other such features, however we focused on the subset that coincides with Stan (continuous variables) for our experiments since these have seen a lot of interest and use among practitioners in recent years.
>
> On a more practical note, our programs must fit into the maximum token length supported by the transformer (512 in our case, though we could switch to LongFormer).
>
> >  It is a little bit difficult for readers with little background knowledge.
>
> We would love to make the discussion around STAN and PPLs more clear. To do so, we have a few more questions on which parts you found difficult.
>
> What would be helpful regarding STAN? Is it a discussion of NUTS or what the STAN programs look like? There are likely too many programs from PosteriorDB to discuss one by one, but we can provide examples of STAN programs in the Appendix if that is helpful?
> Was the discussion of the six small programs in the Appendix helpful? We write them in mathematical notation as well as a text description. What would improve the clarity?
>
> We will try to make further clarifying changes before the final revision!
>
> > High computational cost.
>
> We would like to distinguish between the cost of pretraining the MLI, and finetuning to do inference for a specific program.
>
> The former is very expensive and we agree should be done sparingly. We imagine a large pretrained model (trained on a large corpus of probabilistic programs) being available online for download (similar to how pretrained text models can be downloaded easily off huggingface) in the near future. This would make it so users do not need to perform the pretraining themselves.
>
> The latter, finetuninng, is not as expensive. Figure 5 compares the wall clock time of finetuning (both the gradient steps as well as the forward pass for doing inference) to the wall clock time of HMC, and shows that our approach can be significantly cheaper. While it is true our approach requires a GPU, it is not difficult to imagine finetuning (10 or 100 steps) as an alternative to MCMC or VI.
>
> > What kinds of probabilistic problems justify higher computational cost?
>
> We would argue that programs within the PosteriorDB are representative of the complexity of useful probabilistic programs: they can contain large observation datasets (thousands of entries), a diverse set of function calls, and can have long program lengths – they are similar to probabilistic models used in practice in social sciences, statistics, and business data analytics. For example, the database includes item response theory models to evaluate student ability on exams, GARCH models used in finance, Gaussian mixture models, and timeseries models like ARMA. While there certainly are more complex probabilistic programs, these are already quite rich and useful. We believe a new inference algorithm in this context is meaningful.
>
> That said, one of the exciting aspects of VI for PPLs is scaling to much larger programs or, especially, data sets. We believe foundation posteriors will be a substantial improvement in such cases.

---

> > ### Comment · Reviewer_qzPM · 2022-08-09
> > **Response**
> >
> > Thanks for the reply and I believe most of my concerns have been addressed.
> >
> > In terms of the STAN question, I would suggest adding explanations of what kinds of programs STAN supports and what common features they have. It does not necessarily need to be thorough or formal in the sense that it mainly serves to help authors with limited background to place your work in a bigger literature.

---

### Official Review · Reviewer_KCop · 2022-07-12

**Rating:** 6
**Confidence:** 4
**Soundness:** 3 good
**Presentation:** 3 good
**Contribution:** 2 fair

**Summary:**

  This paper use meta-amortised learning to learn a neural network which can approximate not just a
  family of posterior distributions but also one that is generalised across different probabilistic
  programs. This is done through using compiled inference to generate a corpus of probabilistic programs
  which are then used to fine-tune a transformer model. The language model is trained to predict different
  posterior distributions based on different observations.


**Questions:**

How does this work differ from Che and Yang? What's the contribution of this work in particular?

What makes this work distinct from compiled inference? What approximate inference tasks can this work do that that work struggles with?

**Limitations:**

I feel the authors have adequately addressed the limitations of the work and potential negative social impacts of this work.

**Strengths And Weaknesses:**

  Originality:

  The paper seems very similar to the work of Gwonsoo Che and Hongseok Yang and I'm struggling to
  distinguish them. The main difference seems to be the use of large transformer models instead
  of something that follows the structure of a particular probabilistic program.

  Technical Quality:

  The work mostly focuses on toy problems and the results are promising for them. I have a concern that
  a massive language model is overkill for handling the kinds of toy problems used in this paper. The experiments
  involving Stan and PosterioDB are very compelling though it is a shame that it doesn't out-perform NUTS. I would have
  also liked to see this work compared to the work of Che and Yang from 2021. I'm also concerned that there
  doesn't seem to be any benefit in using an augmented dataset from one toy problem to do inference in
  another toy problem. Without achieving that kind of generalisation, what makes this method better than
  just using compiled inference in each toy domain?

  Clarity:

  The paper is well-written and easy to follow. I had no difficulty understanding the method though
  have not tried to replicate the experiments in the paper.

  Significance:

  I think this work as tremendous potential as being able to do amortised inference over a set of
  related probabilistic programs would be very useful. If this can shown to generalise across several
  domains even related ones it would make the work much more compelling.

---

> ### Author Response · Authors · 2022-08-02
> **Author Response to Reviewer KCop**
>
> Thank you for the helpful questions and comments. We hope to address them below!
>
> > How is this work different from Che and Yang’s approach?
>
> This is a great question and is one that is critical to understanding the novelty of our proposed approach. We enumerate several differences below.
>
> [Our approach trades interpretability for flexibility. Che & Yang (unpublished) make the opposing trade.] For Che and Yang’s approach, there is a single MLP for every atomic operation. For example, there is an MLP specific for sampling, observations, assignment, etc. Given a program (which is an ordering of these atomic operations), the inference network is designed to chain together MLPs in a related ordering. The benefit of this approach is that it is, to a degree, white box since we can at least interpret the inference network as individual pieces, though each piece itself is black-box. There are a few drawbacks to this approach. First, it is not very flexible: suppose we wish to add a new operation (for example, a library function e.g. the Rosenbrock function); we would need to train a new neural network from scratch. Second, it may not be very efficient: for instance, a complex mathematical expression may be written on one or two lines of code but in Che and Yang’s approach may require chaining together a series of MLPs for addition and multiplication to evaluate. And finally, while it is more transparent, it is not clear that Che and Yang’s way of constructing the posterior is the optimal design. In comparison, our approach is fully black box, relying on prediction through a transformer to do inference. However, the benefit is flexibility. A new library function or a new mathematical expression is still at the end of the day, a sequence of tokens, and in this sense requires no change in our approach. Further, given that transformers are quite powerful, our approach is less opinionated: we do not bias the transformer, leaving it to optimization to learn the best approach for inference.
>
> [Our approach generalizes across program structures.] One important contribution of our approach is amortizing across different program structures (generalizing to new unseen programs during training), which Che and Yang do not do (they study a single program structure at a time), meaning they would need to retrain from scratch for every new program. To accomplish this, we introduce program augmentations and show this to be important in achieving generalization. This technique is specific to our approach as well. We emphasize that amortizing inference across program structures is a difficult challenge, and marks a primary distinction between the methods.
>
> [Our approach introduces a new finetuning procedure with VI.] We introduce inference as a two stage procedure: pretraining and transfer, much like a self-supervised learningn setup. During pre-training, we perform a costly procedure to perform “general” performance. During transfer, we use variational inference for finetuning to specialize the posterior for a specific program. This bridges techniques common from NLP to inference. To our best knowledge, this is novel and not used in Che and Yang.
>
> [Our approach performs inference in an orderless autoregressive manner.]  We did not describe this procedure clearly in the original text but now added Section 3.2 to detail the  procedure. To be concrete, we choose a random ordering of the latent variables, unmask one at a time and re-embed the program in between each step: this way, we use inferred values for variables earlier in the ordering when doing inference for variables later in the ordering. This is distinct to the inference in Che and Yang, which is neither orderless nor autoregressive: inference for all m latent variables occurs at once. This distinction means our approach can be a more faithful approximation of the true posterior.
>
> > What makes this work distinct from compiled inference?
>
> One can view our work as a kind of compiled inference (Le et. al. 2017). There are important distinctions though. First, our objective is not only a compiled inference objective; it contains an MLM term which we show in Table 1 to be important for generalization. Second, compiled inference (Le et. al. 2017) does not discuss generalizing across programs for meta-amortized inference; it is traditionally used to perform inference for a single program. Third, using the posterior obtained from our approach as a pre-trained starting point for ADVI is not explored in Le et. al. 2017. And finally, at the very least, our approach shows that similar ideas to compiled inference mix together well with ideas from natural language processing for inference, which we find new and meaningful.

---

> > ### Author Response · Authors · 2022-08-02
> > **Author Response to Reviewer KCop (continued)**
> >
> > > The proposed approach does not outperform NUTS on PosteriorDB programs.
> >
> > We should judge performance of an inference algorithm on two dimensions: the quality of inference and the cost of inference. If we study Figure 5, it is true that the maximum performance of NUTS is higher than our approach. However, this is tautological as the ground truth for PosteriorDB programs is generated by NUTS, so it is not possible to outperform this level in terms of quality.
> >
> > Rather, what we are most excited by is not the maximum performance, but the relationship between cost and performance. In our opinion, Figure 5 demonstrates strongly that our approach outperforms other inference techniques in the low-to-mid cost region (1 to 30 seconds range), whereas HMC requires a large number of mixing steps to achieve good performance, with poor performance before then. This can have important consequences for more complex probabilistic programs. At a certain size HMC is no longer practical while our approach allows the user trade time for performance (fine tuning as long as reasonable).
> >
> > > There doesn't seem to be any benefit in using an augmented dataset from one toy problem to do inference in another toy problem.
> >
> > Perhaps I am misunderstanding the statement but I believe the results in the paper show there to be exactly this kind of generalization. Table 2 shows results when not augmenting to be far worse than with augmenting (-246 LL vs -8 where closer to 0 is better): here, the test set contains a new program not seen before (i.e. not used in training). In Figure 5, we also compare to a baseline (the gray line) where we fit a transformer (of equal parameter count) on each test program, with no pre-training. We can see this approach does not work well compared to pretraining across a dataset of programs, showing a form of a generalization.
> >
> > > A massive language model is overkill for handling the kinds of toy problems.
> >
> > It is true that the transformer model is large, quite possibly larger than needed. However in considering if it is justified, let us distinguish between pre-training and finetuning. Pretraining a transformer is very expensive. In our minds, this is should be done rarely. Much like how large pretrained transformers are open-source today and most ML is done by finetuning them, we imagine there to be a large pretrained model for inference that people can download in the near future. Finetuning a transformer is not that expensive: we showed this in the plots in Figure 5 by plotting wall clock time on the x-axis. For a new probabilistic program, the user needs only to finetune.
> >
> > Furthermore, the programs in PosteriorDB are not considered toy programs by most PPL practitioners. They are certainly not the most complex probabilistic programs but they do contain decently large datasets of observations, have interesting functions and logic, and contain decent program length – they are similar to probabilistic models used in practice in social sciences, statistics, and business data analytics. For example, the database includes item response theory models to evaluate student ability on exams, GARCH models used in finance, Gaussian mixture models, and timeseries models like ARMA. Demonstrating effectiveness on programs of this nature is meaningful.

---

> > > ### Comment · Reviewer_KCop · 2022-08-08
> > > **Re: Response**
> > >
> > > I wish to thank the authors for their detailed and thorough response to my questions. I much better understand the contribution of this work, it's improvements over previous work, and the experimental results that support all that. I have updated my score to reflect this and the further improvements the authors have made to this paper.

---

> > > > ### Author Response · Authors · 2022-08-08
> > > > **Author Response to Reviewer KCop**
> > > >
> > > > Thank you for the response! We are glad that you found the comments helpful.

---

### Official Review · Reviewer_TwoP · 2022-07-12

**Rating:** 8
**Confidence:** 4
**Soundness:** 3 good
**Presentation:** 3 good
**Contribution:** 4 excellent

**Summary:**

Proposes to use BERT style masked language modeling to amortize inference across assignments of variables in probabilistic programs, particularly STAN programs. This learning of an inference network (a across programs is a form of meta-amortization, implemented as compiled inference (optimizing the q(z|x) distribution that maximizes (x,z) ~ p(x,z), using KL(p || q) instead of qp). It allows efficient inference of complex programs, finding high likelihood assignments to variables with low variance in the importance weights. Combined the masked language modeling objective (of random tokens) with masking out just some entries. Attempts to study the learned attention weights to see if the model is learning meaningful causal structure/relations. Performs somewhere between NUTS (best) and ADVI (cheaper) on a dataset drawn from PosterioDB. MLI (masked language inference) does as well as ADVI with a single sample, and improves given more time.

**Questions:**

1. Look into XLNet and orderless NADE. Can decompose the variables in alternate orders and support more flexible decoding.
	2. Wonder how constrained the meta-distribution needs to be to make the problem solvable (see discusssion @ 190)
	3. Consider making the lines in figures have patterns, not just colors (figure 5)
	4. minor edits:
		1. 84: space between "programitself"
		2. Figure A1 is very hard to read on paper
		3. 685: distirbutions -> distributions

**Limitations:**

No significant negative societal impact

**Strengths And Weaknesses:**


	- originality
		- This is a fairly original idea, combining 2 powerful ideas to more effectively amortize and speed up inference in flexible probabilistic models (BERT for PPLs). It is the first paper I am aware of that uses BERT and transformers for inference in probabilistic programs/graphical models. Similar ideas appear in other instances of compiled inference, including applications of LSTMs to learn samplers for sites in probabilistic programs (see Tuan Anh Le thesis and ETAMULIS).
	- quality
		- Well executed, using an established ppl (stan) to evaluate performance. Reports both the likelihood of the sampled values and the variance in importance weights, a helpful diagnostic and indicator of the quality of the approximate posterior. Impressive results compared to standard baselines
	- clarity
		- Well written. Illustrates main points, provides compact background on the motivation for probabilistic programs and approximate inference using compiled inference networks. Believe the results are straightforward to reproduce. Experiments are targeted and make a convincing case for the components of the method and in comparison to standard inference methods built into STAN.
	- significance
		- Suggests a way to use the rapid advances in large language models to aid efficient inference in existing probabilistic programming languages. This is likely to spark significant followup work to learn meta-amortized inference models which are capable of outperforming existing optimizers across a wide range of applications.

---

> ### Author Response · Authors · 2022-08-02
> **Author Response to Reviewer TwoP**
>
> Thank you for the suggestions and positive comments.
>
> > Look into XLNet and Orderless NADE.
>
> Flexible decoding by decomposing the joint into any order of conditions is a great idea. We forgot to include this in the submission, but we do a similar thing. Post-training, when using the transformer as our approximate posterior, we unmask latent variables in random order one-at-a-time. Additionally, the percentage of tokens masked (initialized at 15%) is increased over the training lifetime to cover a larger distribution of inference problems (otherwise, this progressive unmasking technique may be challenging if there are too many masked variables).
>
> As a concrete example, suppose we have the following toy problem:
> ```
> x ~ gaussian(0, 1) -> 0.415
> y ~ gaussian(0, 1) -> -0.123
> a ~ gaussian(x, 1) -> <mask>
> b ~ gaussian(y, 1) -> <mask>
> c = a + b -> <mask>
> ```
> We do not unmask all three variables at once (which is possible to do). Rather, we wish to provide information on intermediate choices made. This requires choosing an order. One potential order to unmask variables is “a”, “b”, then “c” (top-down). Here, we first unmask “a”, building a new program:
> ```
> x ~ gaussian(0, 1) -> 0.415
> y ~ gaussian(0, 1) -> -0.123
> a ~ gaussian(x, 1) -> 0.654
> b ~ gaussian(y, 1) -> <mask>
> c = a + b -> <mask>
> ```
> Then unmask “b”, treating the unmasked value for “a” as an observation. Finally, unmask “c” (now conditioned on observations for “a” and “b”).
>
> However, this order is one of many: we may first unmask “c”, then “a” then “b”. For our implementation, this order is randomly chosen. Given the similarity of XLNet, Orderless NADE, we will add these as references in the related work for the final draft. In particular, XLNet seems like it could be a good backbone architecture to use over RoBERTa for future work. We have included a similar discussion along with a few other missing details in Section 3.2 in the updated rebuttal revision.
>
> > Consider making the lines in figures have patterns, not just colors.
>
> Great idea! We have changed Figure 5 to include different patterns rather than only circles. See rebuttal revision.
>
> > How constrained meta-distribution needs to be to make the problem solvable?
>
> This is a difficult question and I don’t think we have a good answer for it at the moment. Clearly, if the meta-distribution is too wide and the training dataset too small, this meta-inference problem is not solvable. I think the only concrete statement that can be made is that programs of the same style and difficulty within PosteriorDB are constrained enough to make the problem solvable. For longer programs with more difficult logic, a similar approach may be possible but would require a significantly larger dataset to be collected. This remains an open question.

---

### Author Response · Authors · 2022-08-02
**Updated Rebuttal Revision**

We thank all the reviewers for their effort, helpful questions, suggestions, and comments. We uploaded a new rebuttal revision, and provide a summary of the main changes below:

- Added Section 3.2 to provide missing details on using a trained MLI model to do inference through orderless auto-regressive decoding.
- Added a few sentences to Section 5 to make clear that variational finetuning makes the assumption that random variables are continuous.
- Added Section A.1 to describe new potential applications of foundation posteriors to more traditional inference algorithm e.g. MCMC, HMC, and NUTS. One of the main use cases of our approach is to make existing inference algorithms easier to use.
- Added new experiments in Section A.9 for visualizing attention weights rather than norms of contextual vectors.

For each of the reviews below, we respond to specific questions and comments inline. We look forward to discussing any further questions!

---

### Meta-Review · Area_Chair_8fJW · 2022-08-25

**Recommendation:** Accept
**Confidence:** Certain

**Metareview:**

This paper proposes a novel and interesting perspective on leveraging large masked language models as ways to initialize posterior distributions across probabilistic programming language (PPL) tasks. The idea is that this distribution can later be fine tuned over different probabilistic programs.

All reviewers acknowledged that the idea is a novel application of masked language models and, despite being a natural analogy to the way these models are already used nowadays in NLP, can be potentially impactful for amortized inference in PPLs.

The paper is accepted upon the introduction of the discussions emerged during the rebuttal concerning related works and presentation. I also advise the authors to think about substituting the term "Foundation" with a more precise technical term ("Masked Language Model Posteriors"?, "Transformer Posteriors"?,...). Imprecise umbrella terms, nowadays, bring more noise than help and inflate the hype around simple concepts.

**Award:**

No

---

### Decision · Program_Chairs · 2022-09-14

Accept